# Highly Efficient Fluorescent Detection of Vitamin B_12_ Based on the Inner Filter Effect of Dithiol-Functionalized Silver Nanoparticles

**DOI:** 10.3390/nano13172444

**Published:** 2023-08-29

**Authors:** Phan Ba Khanh Chau, Trung Hieu Vu, Moon Il Kim

**Affiliations:** Department of BioNano Technology, Gachon University, Seongnam, Gyeonggi 13120, Republic of Korea; khanh.chau.0767@gmail.com (P.B.K.C.); hieu.vutrung24596@gmail.com (T.H.V.)

**Keywords:** silver nanoparticles, dithiol functionalization, inner filter effect, vitamin B_12_, fluorescent biosensor

## Abstract

We report a fluorescent assay for the determination of vitamin B_12_ (VB_12_) based on the inner filter effect (IFE) of 1,3-propanedithiol-functionalized silver nanoparticles (PDT-AgNPs). PDT was simply functionalized on the surface of AgNPs through Ag–thiol interaction, which leads to significantly enhanced fluorescence, with excitation and emission at 360 and 410 nm, respectively, via their thiol-mediated aggregation. Since target VB_12_ has strong absorption centered at 360 nm, which is almost completely overlapping with the excitation spectra of PDT-AgNPs, the VB_12_ induced strong quenching of the fluorescence of PDT-AgNPs via IFE. The IFE-based mechanism for the fluorescence quenching of PDT-AgNPs in the presence of VB_12_ was confirmed by the analyses of Stern–Volmer plots at different temperatures and fluorescence decay curves. The fluorescence-quenching efficiency of PDT-AgNPs was linearly proportional to the concentration of VB_12_ in a wide range of 1 to 50 μM, with a lower detection limit of 0.5 μM, while preserving excellent selectivity toward target VB_12_ among possible interfering molecules. Furthermore, the PDT-AgNPs-mediated assay succeeded in quantitatively detecting VB_12_ in drug tablets, indicating that PDT-AgNPs can serve as an IFE-based fluorescent probe in pharmaceutical preparations by taking advantages of its ease of use, rapidity, and affordability.

## 1. Introduction

VB_12_, also termed as cobalamin, is one of the water-soluble vitamins belonging to vitamin B family. It is a complex organometallic compound containing a corrin ring coordinated with cobalt ion in the center [1]. In the human body, VB_12_ is recognized an essential cofactor in many metabolic processes, such as red blood cell formation and nerve cell maintenance [2]. In particular, VB_12_ cannot be naturally synthesized in the human body, and, thus, it should be externally supplied from daily food or vitamin supplements. If there is a VB_12_ deficiency, it may lead to serious health issues, such as anemia, gastrointestinal symptoms, neuropathy, ataxia, and cardiovascular diseases [3]. An excess intake of VB_12_ can also cause unexpected adverse effects, such as liver disease, kidney failure, and bone-marrow dysplasia [4]. Therefore, detecting the level of VB_12_ in physiological fluids, food, medicine, and environmental water is of great importance.

There have been several analytical methods to detect VB_12_, such as high-performance liquid chromatography [5], atomic absorption spectroscopy [6], the electrochemical method [7], and the chemiluminescent method [8]. These methods are reliable and selective; however, they generally require high-cost detection equipment and time-consuming sample/electrode preparation steps with skillful operators, which limit their utilization in resource-limited environments.

On the other side, the fluorescent method can provide several superiorities, such as easy operation, rapid detection, and high sensitivity [9]. For the simple procedure without the inclusion of step-by-step washing steps, a target-mediated “turn-on” or “turn-off” fluorescent strategy has commonly been used. To this, diverse fluorescence-quenching mechanisms, such as static and dynamic quenching [10,11], fluorescence resonance energy transfer [12,13], photo-induced electron transfer [14], and IFE [15,16,17,18], have been used. Among them, IFE can be the most convenient one since it can be constructed by just mixing the absorber and fluorophore without any surface modification or chemical conjugation [19]. The IFE can be created when the absorbers’ absorption spectra overlap with the fluorophores’ excitation and/or emission spectra and, thus, the majority of IFE-based detection methods employ a “turn-off” mode, where target molecules function as an absorber to reduce the emission of fluorophores [20]. Although the IFE-based strategy is useful for converting the less sensitive absorption signal of a target into a sensitive fluorescence signal, an effective overlap between the target’s absorption peak and the fluorophore’s excitation and/or emission peak is crucially required. To this, diverse dyes and several fluorescent nanoparticles, including quantum dots, silicon nanoparticles, and noble metal nanoparticles, have been explored for their possible use as efficient fluorophores in IFE-based detection systems [21].

Since AgNPs have a high extinction coefficient and distance-dependent optical characteristics that allow for selective wavelength overlap with the absorber, they have been employed as fluorophores in IFE-based detection systems [22,23]. For example, acetylcholinesterase [22], cyanide ion [24], and ascorbic acid [25] have been identified based on the AgNPs-mediated IFE. However, for more efficient utilizations of AgNPs in the construction of an IFE-based detection system, further improvement of fluorescent characteristics of AgNPs is critically demanded. Herein, we functionalized dithiol-containing ligands, PDT, on the surface of AgNPs, to induce their self-aggregation and concomitant fluorescence enhancement, which are beneficial to serve as fluorophores in the IFE-based detection system. Based on the advantages of PDT-AgNPs, we demonstrated that the aggregated PDT-AgNPs exhibited significantly improved fluorescence, having excitation and emission at 360 and 410 nm, respectively. Importantly, the absorption spectra of VB_12_ nearly perfectly overlapped with the excitation spectra of PDT-AgNPs. Thus, the PDT-AgNPs could serve as a fluorophore for the detection of VB_12_ via IFE. Compared with the existing methods, the PDT-AgNPs-mediated system is convenient to detect VB_12_ by merely mixing them with VB_12_ at room temperature (RT). Right after the mixing, significant fluorescence quenching occurred abruptly and continuously showed the maximal value for 10 min, which is quite beneficial for realizing practical applications.

## 2. Experimental Section

### 2.1. Materials

Silver nitrate, sodium borohydride (NaBH_4_), PDT, glutathione (GSH), L-cysteine (CTY), cysteamine (CTA), 1,4-dithioreitol (DTT), VB_12_ and other vitamins (B_1_, B_2_, B_3_, B_5_, and C), and diverse chloride-based metal salts (CuCl_2_, ZnCl_2_, FeCl_2_, NiCl_2_, MnCl_2_, CeCl_3_, HgCl_2_, NaCl, KCl, CaCl_2_, and MgCl_2_) were purchased from Sigma-Aldrich (St Louis, MO, USA). All solutions were prepared with deionized (DI) water purified by a Milli-Q Purification System (Millipore, Darmstadt, Germany). All chemicals were of analytical grade and used without further purification.

### 2.2. Synthesis and Characterization of AgNPs and PDT-AgNPs

AgNPs were synthesized by the reduction of silver nitrate using NaBH_4_ as a reducing agent. In brief, 12 mg NaBH_4_ were added to a 100 mL aqueous solution containing silver nitrate (0.1 mM), to obtain bright-yellow colored AgNPs during a 5 min in situ reduction at 25 °C, under dark conditions with stirring. To functionalize PDT on the surface of the AgNPs, 2 mL of PDT (1 mM) were added in a dropwise manner into the AgNPs solution, followed by a 2 h reaction at constant stirring. The resulting solution was purified by syringe and centrifugal filters, to collect the precipitates. The obtained suspension of PDT-AgNPs was redispersed in DI water to make a stock solution with a concentration of ~100 pM, determined by the Beer–Lambert law [26]. Other thiol-containing molecules (GSH, CTY, CTA, and DTT) were employed rather than PDT to synthesize the corresponding control thiol-functionalized AgNPs.

Morphologies of AgNPs, PDT-AgNPs, and the mixture of PDT-AgNPs incubated with VB_12_ (PDT-AgNPs/VB_12_) were characterized by transmission electron microscopy (TEM, FEI, Tecnai, OR, USA). For the TEM analysis, the aqueous sample solutions were sonicated for 10 min and dropped into carbon-coated copper TEM grids with air drying overnight. Elemental composition was analyzed by energy-dispersive X-ray spectroscopy (EDS), co-operated with the TEM system. Size distributions of the synthesized materials were examined by a nanoparticle tracking analyzer (NTA) (ZetaView PMX-120, Particle Metrix, Mebane, NC, USA). Surface zeta potentials were analyzed using a Zetasizer Nano-ZS (Malvern Co., Malvern, United Kingdom). A Fourier transform infrared (FT-IR) spectrometer (FT/IR-4600, JASCO, Easton, MD, USA) was utilized to acquire the FT-IR spectra of AgNPs, PDT-AgNPs, and their mixtures with VB_12_. Fluorescence lifetime was examined using a fluorometer (QuantaMaster400, Horiba, Edison, NJ, USA). Quantum yield of PDT-AgNPs was determined by a comparative method with quinine sulfate (quantum yield of 54% in an aqueous solution containing 0.1 M H_2_SO_4_) as a standard reference, using a fluorescence spectrometer (FluoroMate FS-2, Scinco Co., Seoul, Republic of Korea) to record the fluorescence emission spectra at an excitation wavelength of 360 nm.

### 2.3. Fluorescent Detection of VB_12_

Typically, VB_12_ detection was performed by incubating VB_12_ at diverse concentrations (20 μL) with as-synthesized PDT-AgNPs (20 μL) in a sodium phosphate buffer (20 mM, pH 7, 160 μL). After 5 min reaction at RT, the mixture was transferred to a black 96-well plate to measure the fluorescence intensity using a microplate reader (Synergy H1, BioTek, VT, USA), with excitation and emission wavelengths of 360 and 410 nm, respectively. F_0_ and F are the fluorescence intensities in the absence and presence of target VB_12_, respectively, and the F_0_/F was used as the fluorescence-quenching efficiency. Relative F_0_/F (%) was also used, which was calculated by using the ratio of F_0_/F obtained at each sample to the maximal F_0_/F. The limit of detection (LOD) values were calculated according to the equation LOD = 3 S/K, where S is the standard deviation of the blank fluorescence-quenching efficiency and K is the slope of the calibration plot between the concentration of VB_12_ and F_0_/F. For evaluating the selectivity, interfering molecules, such as ions (Cu, Zn, Fe, Ni, Mn, Ce, Hg, Na, K, Ca, and Mg) and vitamins (B_1_, B_2_, B_3_, B_5_, and C), were employed with a 10-fold higher concentration (1 mM) than that of VB_12_ (0.1 mM).

Stabilities of the PDT-AgNPs-mediated system toward VB_12_ were measured under static conditions in ranges of pH (2–10) and storage time of up to 5 days at RT. For pH stability, the PDT-AgNPs were incubated in an aqueous buffer prepared at different pHs for 3 h and, then, their sensing activities were measured under standard conditions, as described above. The storage stability was checked in an aqueous buffer (sodium phosphate, 20 mM, pH 7) for 5 days at RT. The activities measured before the incubation were set as 100% and the relative activity (%) was calculated using the ratio of the residual activity to the initial activity of each sample.

The developed PDT-AgNPs-mediated system was applied to determine the concentrations of VB_12_ in the VB_12_ tablets purchased from a local pharmacy. The drug tablets were pretreated according to the previous protocols [27]. Briefly, the sugar coating on the tablets was removed and they were ground into power. Then, 0.5 g of VB_12_ powder were dissolved in 10 mL DI water. The VB_12_ solution was sonicated for 30 min, filtered by a 0.22 μm filter, and the filtrate was finally diluted to 100 mL to make VB_12_ stock. Then, the VB_12_ stock solution was 1000-fold diluted with water, and predetermined amounts of VB_12_ were added to make spiked samples. PDT-AgNPs were incubated with the appropriately diluted VB_12_ solution and the other procedures were the same as described above. The recovery rate [recovery (%) = measured value/expected value × 100] and the coefficient of variation [CV (%) = SD/average × 100] were determined to assess the precision and reproducibility of the assays.

## 3. Results and Discussion

### 3.1. Synthesis and Characterization of PDT-AgNPs

The overall strategy involved in the synthesis of PDT-AgNPs and the detection method for VB_12_ are represented in Figure 1. AgNPs were first prepared by a conventional reduction of Ag^+^ to Ag^0^ with a NaBH_4_ reducing agent [28]. By the addition of PDT, PDT-AgNPs were prepared with a larger size than that of bare AgNPs and aggregated themselves through an agglomeration phenomenon and Ag–thiol interaction, respectively, possibly resulting in enhanced fluorescence from the alteration of their localized surface plasmon resonance [29]. Importantly, the PDT-AgNPs showed excitation spectra centered at around 360 nm, which was nearly perfectly overlapping with the absorption spectra of VB_12_. Thus, we expect that the PDT-AgNPs can serve as an efficient probe for the fluorescent detection of VB_12_ since the VB_12_ would induce strong quenching of the fluorescence of PDT-AgNPs via IFE. Overall, the PDT-AgNPs-mediated system can be applied to quantitatively determine VB_12_ by simply detecting their fluorescence-quenching signals.

The synthesized AgNPs, PDT-AgNPs, and PDT-AgNPs–VB_12_ were characterized by TEM and NTA. Yellow-colored bare AgNPs were in a monodispersed state having uniform size (3.2 ± 0.2 nm) (Figure 2a,d). PDT-AgNPs showed a brown color and, as expected, they were larger and in an aggregated state having an average size of 13.3 ± 0.1 nm (Figure 2b,d), confirming their enlargement and self-aggregation via agglomeration and Ag–thiol interaction, respectively [27,30]. Importantly, by the incubation with VB_12_, the size of the NPs was marginally enlarged up to 14.3 ± 1.7 nm (Figure 2c,d). This was presumably due to the coordination of Co^2+^, the element located at the center of VB_12_, with the thiol groups of PDT-AgNPs through metal–ligand interaction [30]. Additionally, the quantum yield of PDT-AgNPs was determined to be 3.3%, which is similar to the reported value of bare AgNPs (Appendix A) [31,32]. To validate the successful conjugation of PDT on AgNPs, FT-IR and EDS analyses were additionally performed. The FT-IR spectra clearly support the presence of thiol-containing PDT in PDT-AgNPs and corrin ring-containing VB_12_ in PDT-AgNPs–VB_12_ (Appendix A). The peaks at around 500–600 cm^−1^ and 570–700 cm^−1^, which corresponded to Ag-S bonds and S-S bonds, respectively, were observed in both PDT-AgNPs and PDT-AgNPs–VB_12_, while only PDT and VB_12_ did not show the peaks due to the absence of chemical groups [33,34]. The peaks observed at 1670 cm^−1^ may indicate the corrin ring of VB_12_ [35], which were observed in PDT-AgNPs–VB_12_, although the peak intensity was relatively low due to the noise signals [36,37]. EDS analysis further provides evidence of the elemental composition of PDT-AgNPs having two major elements of Ag and S, while bare AgNPs did not show the peaks corresponding to the S element (Appendix A). Notably, PDT-AgNPs showed a large increase in surface zeta potentials (−9.27 ± 0.40 mV) when compared with bare AgNPs (−21.08 ± 0.43 mV), supporting the successful conjugation of neutrally charged PDT ligands on the surface of AgNPs (Appendix A) [38,39]. These characterizations confirmed that PDT-AgNPs were successfully formed in a self-aggregated state, with preserving thiol groups on their surface, which induced further aggregation in the presence of VB_12_.

### 3.2. IFE-Based Fluorescence Quenching of PDT-AgNPs in the Presence of VB_12_

PDT-AgNPs are expected to have enhanced fluorescence due to the alteration of their localized surface plasmon resonance, via the growth of particles during PDT functionalization and thiol-mediated aggregation, as discussed in previous reports [29,30]. During the surface functionalization, the thiol groups of PDT are prone to capture the surfaces of AgNPs by forming linking bridges (Ag-S and S-S), leading to particle enlargement by agglomeration phenomenon as well as self-aggregation [29], which may trigger the fluorescence enhancement of PDT-AgNPs [30]. To examine this, the fluorogenic behaviors of AgNPs and PDT-AgNPs were compared. The investigation clearly showed that the excitation and emission peak intensities of PDT-AgNPs were over five- and twofold greater than those of bare AgNPs, respectively, as shown using the arrows in Appendix A, which is highly beneficial to creating a fluorescence-based detection system. Moreover, the excitation spectra of PDT-AgNPs, located at around 360 nm, almost overlapped with the absorption spectra of VB_12_ (Figure 3a), which demonstrates that VB_12_ can lead to the attenuation of fluorescence emission of PDT-AgNPs, possibly via IFE. The excitation spectra of PDT-AgNPs displayed a proportional decrease as the concentrations of VB_12_ increased (Appendix A), as expected. Next, the fluorescence-quenching efficiencies (F_0_/F) of bare AgNPs, PDT-AgNPs, and other control thiols-functionalized AgNPs, such as GSH, CTY, CTA, and DTT, in the presence of target VB_12_, were compared. The results showed that the F_0_/F value of PDT-AgNPs was nearly threefold higher than those of bare and other thiols-functionalized AgNPs (Figure 3b), proving the essential role of PDT in the fluorescence quenching performed toward VB_12_. It is presumably due to the excitation spectra of other thiols-functionalized AgNPs (GSH-AgNPs ~400 nm, CTY-AgNPs ~395–417 nm, CTA-AgNPs ~400 nm, and DTT-AgNPs ~408 nm), which were apart from the absorption spectra of VB_12_ [40,41,42,43].

Fluorescence quenching can be caused by several different mechanisms and the distinct feature of IFE can be certified by analyzing Stern–Volmer plots at different temperatures and fluorescence decay curves. The Stern–Volmer equation was presented as F_0_/F = 1 + K_Q_τ_0_[Q] = 1 + K_SV_[Q] [44], where F0 and F are the fluorescence intensities in the absence and presence of an absorber (VB_12_ in this study), respectively, [Q] is the concentration of VB_12_, K_SV_ is the Stern–Volmer quenching constant, K_Q_ is the quenching rate coefficient, and τ_0_ is the lifetime of the fluorophore (PDT-AgNPs in this study). Since IFE is not temperature dependent, which is obviously different from other quenching mechanisms such as static and dynamic quenching [9], we investigated the influences of several temperatures (298, 308, and 318 K) on the quenching efficiency toward VB_12_. As a result, linear curves were plotted based on the relationship between F_0_/F and concentrations of VB_12_, but the slope values were not temperature dependent (Figure 3c). In addition, the quenching constants obtained at different temperatures, calculated by the Stern–Volmer equation presented above, were nearly the same (Appendix A). These results indicate that the quenching mechanism was possibly IFE.

Moreover, fluorescence decays were measured to clarify the quenching mechanism, because the fluorescence lifetime of the fluorophore is almost constant in IFE-based quenching [7]. The experiments showed that the fluorescence lifetime of PDT-AgNPs was almost unchanged before and after adding VB_12_ (Figure 3d), confirming that the mechanism governing the fluorescence quenching of PDT-AgNPs in the presence of VB_12_ is IFE.

### 3.3. Fluorescent Detection of VB_12_ Using PDT-AgNPs-Mediated Detection System

Via the IFE-based fluorescence quenching of PDT-AgNPs, the target VB_12_ was quantitatively determined by detecting the increased fluorescence-quenching efficiency (F_0_/F) as the concentrations of VB_12_ increased. Several parameters affecting the fluorescence quenching, such as PDT concentration, pH, temperature, and reaction time, were examined to set up the optimal reaction conditions (Appendix A). One mM PDT yielded the best quenching efficiency than the other values of PDT to prepare PDT-AgNPs. PDT-AgNPs showed relatively higher quenching efficiency (F_0_/F > 3) in a broad range of pH (3~10) and ambient temperature (20~35 °C). Moreover, the F_0_/F value abruptly increased and reached close to its maximal value right after mixing PDT-AgNPs and VB_12_, and continuously showed the value for 10 min. Considering the fluorescence-quenching efficiency and experimental convenience, 1 mM PDT, pH 7.0, RT, and 5 min reaction time were adopted for the detection of VB_12_ in further studies.

Under the optimized conditions, target VB_12_ was determined via PDT-AgNPs-mediated IFE-based assay. Through the simple 5-min incubation at RT, VB_12_ was selectively determined by the vivid fluorescence quenching (Figure 4a,b and Appendix A). On the other hand, interfering molecules, such as ions (Cu, Zn, Fe, Ni, Mn, Ce, Hg, Na, K, Ca, and Mg) and vitamins (B_1_, B_2_, B_3_, B_5_, and C) did not yield any considerable quenching signal, where they are used at 10-fold higher concentrations. These results demonstrate that a specific fluorescence quenching occurred by a PDT-AgNPs-mediated reaction only from target VB_12_. From the fluorescence spectra with the analysis of the calibration plot, LOD was determined to be as low as 0.5 µM with a linear range from 1 to 50 µM (Figure 4c,d). These LOD and linear range values are comparable with the best results of those in recent reports describing VB_12_ detection and enough to practically determine the VB_12_ level in pharmaceuticals (Table 1). Compared with the reported methods, the PDT-AgNPs-mediated system is more rapid and convenient to perform and, furthermore, the highest level of detection sensitivity was achieved.

Furthermore, the stabilities of the PDT-AgNPs-mediated system were evaluated over different pHs and storage times (Appendix A). The experimental results clearly showed that the sensing activity of the PDT-AgNPs toward VB_12_ did not show any significant change with the alterations of reaction pH (2–10) and storage time up to 5 days at RT. This finding confirms the robust nature of PDT-AgNPs, which are beneficial to realize their applications in the field.

Finally, to investigate the practical applicability, the PDT-AgNPs-mediated detection system was utilized to quantify the concentrations of VB12 in drug tablets. The VB12 drug was dissolved in DI water at an appropriate level and two different amounts of VB12 were added to make spiked samples. As a result, the VB12 formulations were precisely and accurately determined, yielding CVs in a range of 2.2–3.7% and recovery rates of 96.5-102.4% (Table 2). These results demonstrate that the developed PDT-AgNPs-mediated IFE-based detection system could serve as a convenient and reliable analytical tool for the analysis of VB12 in pharmaceutical formulations.

## 4. Conclusions

We constructed an IFE-based “turn-off” fluorescent detection method for VB_12_ using PDT-AgNPs. PDT functionalization on AgNPs induced highly enhanced fluorescence intensities by inducing their aggregation and, importantly, the excitation spectra of PDT-AgNPs completely overlapped with the absorption spectra of VB_12_, enabling IFE-based fluorescent VB_12_ detection. Based on the phenomena, VB_12_ could be determined with high selectivity and sensitivity, along with its successful quantification in drug tablets with sufficient detection precisions. This study provides an efficient way to develop highly efficient fluorescent probes with simple surface engineering and IFE-based biosensors, providing significant potential for being used in VB_12_ detection in pharmaceutical preparations.

## Figures and Tables

**Figure 1 nanomaterials-13-02444-f001:**
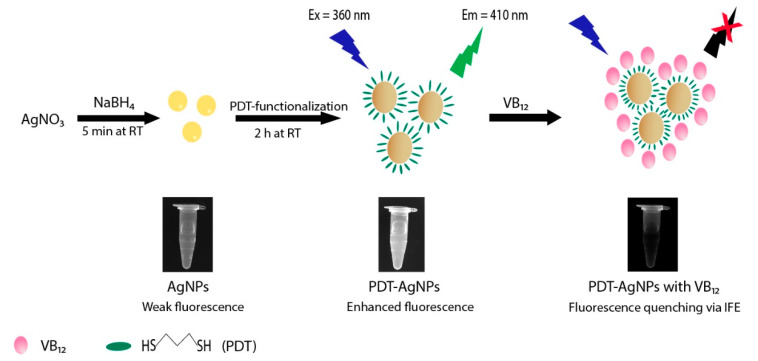
Schematic illustration for the synthesis of PDT-AgNPs and their application in fluorescent detection of VB_12_ via IFE.

**Figure 2 nanomaterials-13-02444-f002:**
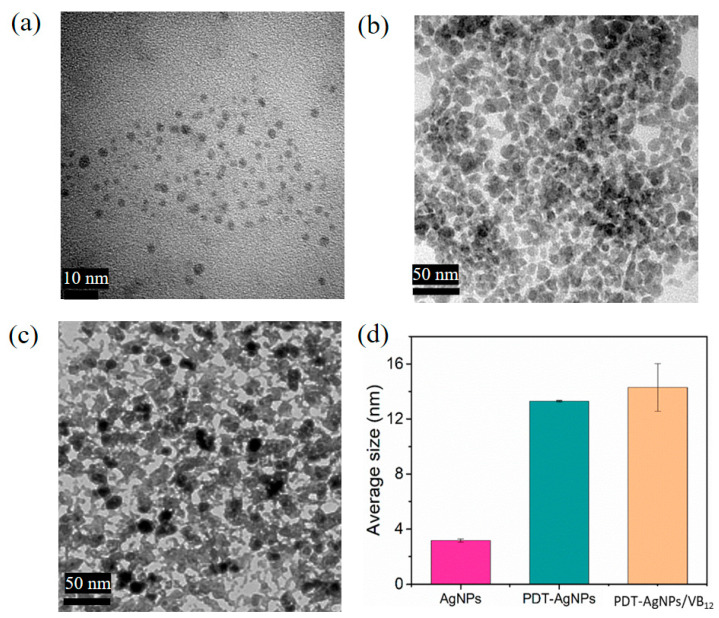
TEM images of (**a**) bare AgNPs, (**b**) PDT-AgNPs, and (**c**) PDT-AgNPs–VB_12_, with their (**d**) size distributions.

**Figure 3 nanomaterials-13-02444-f003:**
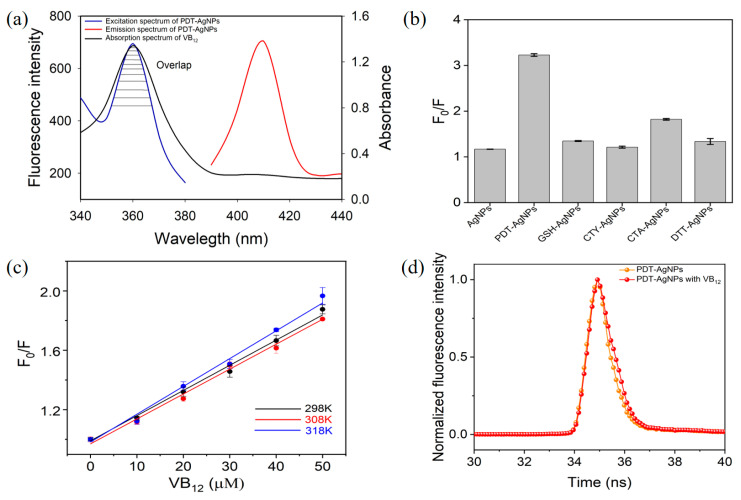
Quenching mechanism of PDT-AgNPs for the detection of VB_12_. (**a**) Overlap between absorption spectra of VB_12_ and excitation spectra of PDT-AgNPs. (**b**) Fluorescence-quenching efficiencies (F_0_/F) of bare AgNPs, PDT-AgNPs, and other thiols-functionalized AgNPs. (**c**) Stern-Volmer plots at different temperatures. (**d**) Fluorescence decay of PDT-AgNPs with and without VB_12_.

**Figure 4 nanomaterials-13-02444-f004:**
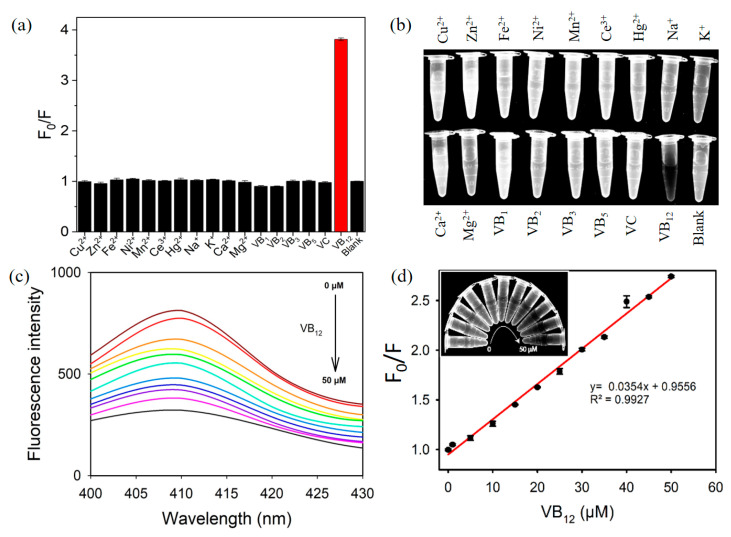
Analytical capability of PDT-AgNPs-mediated IFE-based assay. (**a**,**b**) Selectivity toward VB_12_ with its corresponding images. (**c**,**d**) Fluorescence spectra and their corresponding linear calibration plot for the determination of VB_12_. Colored lines in (**c**) indicate the changes of the concentration of VB_12_.

**Table 1 nanomaterials-13-02444-t001:** Comparison of linear range and LOD values of the PDT-AgNPs-mediated detection system with those of recent reports describing VB_12_ detection.

Method	Material	Linear Range (μM)	LOD (μM)	Reference
HPLC		10.5–28	0.93	[5]
Atomic spectroscopy		2.6–66.6	0.9	[6]
Electrochemistry	Mn-DNA-CPE	3.667–236	1.21	[7]
Colorimetric assay	Nitroso-R-salt	0–12.25	Not given	[45]
Fluorescent assay	Hydroxypropyl-β-cyclodextrin	0–21	0.18	[46]
Fluorescent assay (FRET)	CdTe quantum dots	0.7–9.8	0.1	[12]
Fluorescent assay (FRET)	CdS quantum dots	3.7–74	5.1	[13]
Fluorescent assay (IFE)	Nitrogen-doped carbon dots	0–200	2.045	[47]
Fluorescent assay (IFE)	Carbon dots	1–65	0.62	[16]
Fluorescent assay (IFE)	Carbon quantum dots	0.75–100	0.2	[18]
Fluorescent assay (IFE)	Carbon dots	0–60	0.1	[15]
Fluorescent assay (IFE)	PDT-AgNPs	1–50	0.5	This work

**Table 2 nanomaterials-13-02444-t002:** Detection precision of PDT-AgNPs-mediated assay for the quantitative determination of VB_12_ in drug tablets.

Compound	Spiked Level (µM)	Measured (µM) ^a^	Recovery (%) ^b^	CV(%) ^c^
VB_12_ tablet	0	3.56	96.50	3.37
	10	13.52	98.77	3.70
	20	24.24	102.33	2.23

^a^ Mean of three independent measurements. ^b^ Measured value/expected value × 100. ^c^ Coefficient of variation.

## Data Availability

The data presented in this study are available on request from the corresponding author.

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
