# Peer review of "Highly Efficient Fluorescent Detection of Vitamin B12 Based on the Inner Filter Effect of Dithiol-Functionalized Silver Nanoparticles"

_nanomaterials, 2023, doi:10.3390/nano13172444_

Round 1

Reviewer 1 Report

Comments to the Author: This manuscript describes a fluorescent sensing system that detects vitamin B12 on the basis of the inner filter effect using the dithiol-functionalized silver nanoparticles. According to our careful evaluation, some details of the experiment need to be perfected. After major revisions, it can be published on Nanomaterials.

1.     The author should explain the reason for the enhanced fluorescence of PDT-AgNPs.

2.     To better confirm the successful conjugation of PDT on AgNPs, please supplement the zeta potentials of AgNPs and PDT-AgNPs.

3.     Some figures need to be improved. For example, “Fluorescence” should be “Fluorescence intensity”. Besides, there are some typos needed to be corrected. For instance, “F0/F” should be “F0/F”.

4.     In order to further illustrate the optical properties of PDT-AgNPs, the fluorescence quantum yield should be provided.

5.     What does “relative F0/F” in Figure S4a mean? How to calculate?

6.     The relevant literatures should be cited, such as, ACS Appl. Mater. Interfaces 2021, 13(42), 50228–50235, Green Chem., 2016, 18 (19), 5127–5132, New J. Chem. 2015, 39(2), 1295–1300.

7.     In order to explore the stability of PDT-AgNPs, the author should supply the influence of different pH and time on the fluorescence intensity of PDT-AgNPs.

8.     In Table 1, the author should give more literature on the detection of VB12 by different methods mentioned in the introduction.

Reviewer 2 Report

The authors presented a method for the detection of VB12 based on functionalized silver nanoparticles. The authors found the fluorescence quenching efficiency was linearly proportional to the concentration of VB12, which can be used for the quantitative detection of VB12. The manuscript was well written and the experimental results and discussions were clearly illustrated. Some suggestions were presented, which may improve the manuscript.

1. The novelty of the research must be clearly claimed and highlighted. The novelty of presented method and results is not enough interesting for readers.

2. The contrast of Figure 4b is relatively low. Please improve it.

3. In Figure 1, the PDT-AgNPs is brighter than AgNPs. What’s the absorpion of AgNPs and PDT-AgNPs.

Reviewer 3 Report

In this manuscript, the authors reported a “turn-off” fluorescent probe based on the inner filter effect for the detection of VB12 with 1,3-propanedithiol-functionalized silver nanoparticles (PDT-AgNPs). The absorption spectrum of VB12 overlaps with the excitation spectrum of PDT-AgNPs, resulting in the quenching of the fluorescence of PDT-AgNPs, which is proportional to the concentration of VB12. Finally, the PDT-AgNPs successfully achieved the quantitative detection of VB12 in drug tablets. I recommend publication of this manuscript in Nanomaterials after major revision. The issues and questions are listed below:

1.The author summarized recent reports on the detection of VB12 in Table 1. Please discuss the advantages of PDT-AgNPs-mediated assay compared to other works.

2. The background of the TEM images are not clean. The size of the PDT-AgNPs is much larger than the bare AgNPs. From the TEM images, it can be clearly observed that the PDT-AgNPs has larger size with solid edge. It seems the enlargement is not due to the aggregation of smaller particles, but the growth of the nanoparticles.

3. In Fig.3b, why does the PDT-AgNPs exhibit the highest fluorescence quenching efficiency compared to other thiol-functionalized AgNPs (GSH, CTY, CTA, and DTT) in the presence of target VB12? Please explain the mechanism.

4. How does the absorption spectrum of PDT-AgNPs change as the concentration of VB12 increases? Relevant experimental resutls should be supplemented.

5. The anti-interfering assessment of the PDT-AgNPs should be provided using the potential interfering substances from the real samples.

6. The baseline of the absorption, excitation and fluorescence spectra in Fig 3a and S3 is drifting. Please correct.

The overall English language is OK for publication.

Round 2

Reviewer 1 Report

Can be accepted in current form.

Reviewer 2 Report

The manuscript was carefully revised and all questions were answered. I suggest to accept the manuscript.

Reviewer 3 Report

The authors have well-addressed the comments raised by the reviewer, thus I recommend publication of this manuscript without further revision.